# A Live Probiotic Vaccine Prototype Based on Conserved Influenza a Virus Antigens Protect Mice against Lethal Influenza Virus Infection

**DOI:** 10.3390/biomedicines9111515

**Published:** 2021-10-21

**Authors:** Daria Mezhenskaya, Irina Isakova-Sivak, Tatiana Gupalova, Elena Bormotova, Eugenia Kuleshevich, Tatiana Kramskaya, Galina Leontieva, Larisa Rudenko, Alexander Suvorov

**Affiliations:** Scientific and Educational Center “Molecular Bases of Interaction of Microorganisms and Human” of the World-Class Research Center “Center for Personalized Medicine” FSBSI “IEM”, 197376 Saint Petersburg, Russia; isakova.sivak@iemspb.ru (I.I.-S.); tvgupalova@rambler.ru (T.G.); bormotovae@rambler.ru (E.B.); k-zh-v@mail.ru (E.K.); tatyana.kramskaya@gmail.com (T.K.); galeonte@yandex.ru (G.L.); vaccine@mail.ru (L.R.); alexander_suvorov1@hotmail.com (A.S.)

**Keywords:** *Enterococcus faecium* L3, influenza, LAH antigen, M2e antigen, IgG, oral immunization, universal influenza vaccine, mouse model

## Abstract

Background: Due to the highly variable nature of the antigenic properties of the influenza virus, many efforts have been made to develop broadly reactive influenza vaccines. Various vaccine platforms have been explored to deliver conserved viral antigens to the target cells to induce cross-reactive immune responses. Here, we assessed the feasibility of using *Enterococcus faecium* L3 as a bacterial vector for oral immunization against influenza virus. Methods: we generated two vaccine prototypes by inserting full-length HA2 (L3-HA2) protein or its long alpha helix (LAH) domain in combination with four M2e tandem repeats (L3-LAH+M2e) into genome of *E.faecium* L3 probiotic strain. The immunogenicity and protective potential of these oral vaccines were assessed in a lethal challenge model in BALB/c mice. Results: as expected, both vaccine prototypes induced HA stem-targeting antibodies, whereas only L3-LAH+4M2e vaccine induced M2e-specific antibody. The L3-HA2 vaccine partially protected mice against lethal challenge with two H1N1 heterologous viruses, while 100% of animals in the L3-LAH+4M2e vaccine group survived in both challenge experiments, and there was significant protection against weight loss in this group, compared to the L3 vector-immunized control mice. Conclusions: the recombinant enterococcal strain L3-LAH+4M2e can be considered as a promising live probiotic vaccine candidate for influenza prevention and warrants further evaluation in relevant pre-clinical models.

## 1. Introduction

The composition of influenza vaccines is updated almost every year to account for the high variability of influenza viruses. Preparing influenza strains for an upcoming influenza season is a very complex process which involves the coordinated collection and analysis of a large number of influenza isolates from multiple WHO centers around the world. The effectiveness of traditional seasonal influenza vaccines may be affected by various circumstances. It is not unusual that circulating influenza viruses differ significantly from vaccine strains in antigenic properties [1,2], or that mutations occur during vaccine strain preparation [3]. Furthermore, licensed influenza vaccines typically induce neutralizing antibodies targeted at immunodominant, but highly variable sites in the globular head domain of the influenza hemagglutinin (HA), making it possible for the virus to easily escape these antibodies [4,5]. Therefore, great efforts have been made over the last decade to develop a universal influenza vaccine, that is, a vaccine with a broad spectrum of action capable of providing long-lasting protection, not only against drifted type A virus strains of one subtype, but from viruses belonging to different subtypes as well [6]. To broaden the spectrum of protective action of influenza vaccines, the immune response should be targeted at viral antigens that are conserved among different subtypes of the influenza A virus [7,8]. Thus, HA2 subunit of viral hemagglutinin is significantly more conserved than its globular HA1 subunit because it functions as viral fusion peptide and anchors the globular head domain to the viral membrane [9]. Furthermore, the long alpha helix (LAH) from the HA stalk domain is highly conserved among influenza virus subtypes belonging to the same evolutionary group [6,10]. Another highly conserved viral antigen, the extracellular domain of the matrix protein 2 (M2e), is also considered a very promising target for universal influenza vaccine design [11], however both LAH and M2e are rather weak immunogens, therefore, various strategies have been explored to increase their immunogenicity [6]. In the current study we generated two live probiotic influenza vaccine prototypes based on *Enterococcus faecium* L3 expressing on their pili either HA2 protein of influenza H1N1 virus or the LAH antigen in combination with tandem M2e repeats and evaluated their protective effect in a lethal challenge model in mice.

## 2. Materials and Methods

### 2.1. Viruses, Proteins and Peptides

The mouse-adapted pandemic influenza virus strain A/California/7/2009 (H1N1) (Cal09 MA) was obtained from Smorodintsev Research Institute of Influenza (St. Petersburg, Russia). A PR8-based reassortant influenza virus encoding M gene of swine origin was generated earlier by means of reverse genetics [12]. The viruses were grown in developing chicken embryos incubated at 37 °С. Allantoic fluid containing the virus was harvested, clarified by low-speed centrifugation, and stored in aliquots at −70 °С. The infectious virus titer was quantified by endpoint dilution assay in chicken embryos and was expressed as 50% egg infective doses (lg EID_50_/mL).

Recombinant chimeric cH6/1 protein consisting of HA head domain from H6N1 A/mallard/Sweden/81/2002 virus and stalk domain from Cal09 virus was kindly provided by Professor F. Krammer (ISMMS, New York, NY, USA). The recombinant 3M2e protein consisting of three consecutive M2e peptides 24 amino acids long each, as well as the recombinant LAH protein, a long alpha spiral of the HA stalk domain of the Cal09 virus (amino acids residues 52-132 of HA2), were kindly provided by Doctor A. Kazaks (Latvian Biomedical Research and Study Centre, Riga, Latvia) [13].

### 2.2. Generation of Enterococcus faecium Encoding Influenza Virus Fragments

We designed two vaccine prototypes based either on a full-length HA2 subunit of influenza A/South Africa/3626/2013 (H1N1) (SA H1N1) virus, or on a combination of conserved antigens LAH and 4M2e. For the construction of HA2 or LAH+4M2e vaccine inserts, the HA2 lacking the transmembrane domain (aa 1 to 178) and the LAH (HA2 aa residues 55 to 128) fragments were amplified directly from viral RNA extracted from SA H1N1 influenza virus using fragment-specific primers. The 4M2e fragment was amplified from a plasmid DNA encoding a chimeric HA+4M2e gene described earlier [14], which includes M2e sequences of influenza A viruses belonging to avian/swine, human/swine, swine, and human lineages, separated by flexible linkers (117 aa long). An overlapping PCR was performed to generate the LAH+4M2e combined gene. The amplified HA2 and LAH+4M2e DNA fragments, both flanked by the sites for NdeI and EcoRI restriction enzymes, were further cloned into the pJET1.2 plasmid using the Clone JET™ PCR Cloning Kit (ThermoFisher, Waltham, MA, USA).

The plasmid pEntF-PspF encoding pneumococcal surface protein F (PspF) within the *d2* gene from *E. faecium* L3- encoding pili protein [15] was hydrolyzed by the NdeI and EcoRI enzymes to remove the PspF gene, and the remaining fragment was used for subsequent cloning of viral epitopes. The digested pJET1.2-(HA2), pJET1.2-(LAH+4M2e) and pEntF-PspF plasmid DNAs were ligated and further transformed into the heterologous system *E. coli* DH5α using selective medium containing 500 μg/mL of erythromycin. The selected integrated plasmids pEntF-HA2 and pEntF-LAH+4M2e were extracted using QIAGEN Plasmid Maxi Kit (Hilden, Germany), and their sequences were confirmed by Sanger sequencing.

The *Enterococcus faecium* L3 culture was transformed by the integrated plasmids using electroporation procedure as described earlier [15]. The resulting transformants were screened by standard PCR with primers specific to bacterial gene and the viral fragments, followed by amplification of the whole integrated fragments with primers annealing to the bacterial chromosomal gene d2-1 (forward, GCTCTAGAGCCGATGAGAGCAGCTGGTATTG; reverse, CAACAGGATCCAAAGCATCGTTGG). These amplified fragments were extracted from agarose gel and subjected to Sanger sequences to confirm the identity of inserted viral antigens (see Appendix A).

The growth properties of *Enterococcus faecium* L3 and its modified clones were evaluated by comparing individual growth curves. For this, 20 mL of liquid THB medium was inoculated with an equal amount of each bacterial strain and incubated for 24 h at 37 °C. At zero hour, and then with an interval of every 1 h for 8 h and at 24 h, the samples of the culture suspensions were taken to count the number of bacteria on agar plates as described above.

### 2.3. Immunization of Mice with Live Probiotic Vaccine and Assessment of Its Immunogenicity

Eight-to-ten-weeks-old female BALB/c mice were purchased from the laboratory-breeding nursery of the Russian Academy of Sciences (Rappolovo, Leningrad Region, Russia) and maintained under standard conditions. All animals were fed by autoclaved food and had access to water ad libitum. As the inbred animals share an identical genetic background, no specific randomization and blinding procedures were performed to divide them into study groups.

Groups of ten mice were administered with 3 × 10^9^ CFU of live probiotic recombinant vaccines into the esophagus in a volume of 300 μL using a mouse gavage needle. The control group received an equal amount of the intact *Enterococcus faecium* L3 probiotic by the same route of administration. This type of control was required to exclude the non-specific antiviral effect of the enterococcal strain possibly mediated by the enterocin B peptide [16]. A vaccine course included daily oral administration of bacteria for three consecutive days. This vaccine course was repeated three times at two-week intervals (Figure 1).

Retro-orbital sinus blood samples were collected from five mice in each group two weeks after the last feeding. This sample size gave an 78% power to detect significant differences in the immunogenicity (i.e., AUC values 3 and 1.1) between study groups with an alpha 0.05 (G*Power version 3.0.10. software, Dusseldorf, Germany). All serum samples were treated with *E. faecium* L3 suspension at 56 °C for 1 h, followed by centrifugation at 3000× *g* for 1 min to remove the bacteria. This treatment with L3 was required to eliminate non-specific binding of anti-L3 IgG antibody with influenza virus antigens.

ELISA was conducted following standard procedures. Briefly, 96-well plates (Greiner Bio-One, Frickenhausen, Germany) were coated with 50 ng/well of either cH6/1, LAH or 3M2e proteins in a carbonate-bicarbonate buffer, in a volume of 50 µL per well and stored at 4 °C overnight. The plates were washed by 0.05% Tween20 in PBS (PBST), then blocked in 50 μL of PBS containing 1% bovine serum albumin (BSA) for 30 min at 37 °C. Serial two-fold serum dilutions were prepared starting from 1:10 and added to the coated wells. The plates were incubated for 1 h at 37 °C, then washed four times with PBST. Following incubation with goat anti-mouse IgG conjugated to horseradish peroxidase (BioRad, Hercules, CA, USA) for 1 h at 37 °C, the plates were washed four times in PBST, and the antibody binding was detected with 3,3′,5,5′-Tetramethylbenzidine substrate (1-Step Ultra TMB–ELISA Substrate Solution, Thermo Fisher Scientific, Waltham, MA, USA). Optical density was measured at 450 nm using xMark Microplate Spectrophotometer (BioRad, Hercules, CA, USA). The area under the curve (AUC) of the OD_450_ values for all dilutions of each individual serum sample was calculated using the trapezoidal rule and expressed in arbitrary units.

### 2.4. Assessment of Protection against Influenza Virus Challenge

Three weeks after the last vaccination, five mice from each immunization group were challenged with 3 LD_50_ of either mouse-adapted A/California/7/09 strain or a recombinant A/PR8-based virus carrying artificial M gene of swine-origin lineage—PR8 (M2sw) [12]. Body weight loss and survival rates were monitored daily for 14 days. Mice that lost more than 30% of their initial body weight were considered dead.

### 2.5. Statistical Analyses

Data were analyzed with the statistical module of GraphPad Prism 6 software. Statistically significant differences in the immunogenicity outcomes (AUC of OD_450_ values) and pathological outcomes (AUC of weight loss values) between study groups were determined by ANOVA with Tukey’s multiple comparison test. Differences in the survival rates after challenge were analyzed by a log-rank Mantel–Cox test. *p* values of <0.05 were considered significant.

## 3. Results

### 3.1. Generation and Characterization of Enterococcal Vaccines

Two live probiotic influenza vaccine candidates were generated by standard gene engineering approaches. The two chimeric DNA constructs containing the HA2 or LAH+4M2e element were inserted into the *d2* Enterococcus gene resulting in the exposure of the influenza virus antigens on the surface of bacteria as a part of their pili (Figure 2A,B). Amplification of the *d2* bacterial gene fragment demonstrated that the desired influenza genes were successfully inserted into bacterial genome (Figure 2C). Importantly, the modification of the L3 enterococci did not affect growth properties of the bacterial strain, suggesting their feasibility for the large-scale probiotic vaccine production (Figure 2D).

### 3.2. Immunogenicity of the Live Probiotic Influenza Vaccine Candidates in Mice

As shown on Figure 3A,B, both L3-HA2 and L3-LAH+4M2e vaccines induced similarly high levels of antibodies targeting the HA stem domain (either in the full-length mode or its conserved LAH domain), whereas the control L3 strain did not induce HA-binding IgG antibody. In turn, only L3-LAH+4M2e vaccine induced detectable levels of anti-M2e antibodies, while the L3 and L3-HA2 vaccines failed to raise antibodies to this conserved viral antigen (Figure 3B).

### 3.3. Protection against Lethal Influenza Virus Infection

To assess the protective effect of induced anti-HA and anti-M2e antibodies, immunized mice were challenged with two H1N1 lethal influenza A viruses, at a dose 3 LD_50_. These challenge viruses were previously described as the source of M genes belonging to different evolutionary lineages, with established lethality for BALB/c mice [12].

Both Cal MA and PR8 (M2sw) viruses resulted in severe weight loss in mice from the control L3 group, with 100% mortality (Figure 4). In contrast, the L3-HA2-immunized animals were protected from lethality after PR8 (M2sw) challenge, although there was no significant difference in the AUC of weight loss parameter between the L3 and L3-HA2 groups (Figure 4B). The L3-HA2 vaccine also partially protected mice against mortality induced by Cal MA, however the difference in the survival rates with the L3 group did not reach statistical significance (*p* = 0.13). Strikingly, a combination of the LAH antigen and M2e tandem repeats were significantly more effective than the HA2 antigen alone: the L3-LAH+M2e candidate protected mice against mortality and weight loss in both challenge experiments, suggesting that the induced HA- and M2e-specific antibodies had a synergistic protective effect (Figure 4).

## 4. Discussion

The development of universal influenza vaccines includes the optimization of existing prototypes, as well as the search for new vectors that can deliver desired conserved influenza virus antigens to the target cells. Here, we report for the first time the development of a more broadly protective influenza vaccine based on a live oral probiotic vaccine platform. We generated two probiotic vaccine candidates expressing either HA2 hemagglutinin subunit of H1N1 influenza virus or its conserved part, LAH antigen, in combination with four conserved M2e epitopes. These antigens were inserted into the d2 *Enterococcus* gene resulting in their exposure on the surface of bacteria as a part of their pili, without any negative effect on the bacterial growth properties. Oral vaccination of mice with both vaccine prototypes induced systemic humoral immune responses to the virus inserts, as evidenced by increased levels of circulating HA- and M2e-specific IgG antibodies in those groups that contained corresponding antigens in the vaccine. Interestingly, the anti-HA antibody only partially protected mice against mortality caused by antigenically diverged H1N1 influenza viruses, and the addition of M2e conserved antigen to the probiotic vaccine was required to achieve statistically significant protection against mortality and weight loss in both challenge experiments. This strategy has been used earlier for the development of a mucosal vaccine against *S. pneumoniae* infection, demonstrating the versatility of the live probiotic vaccine platform for developing vaccines against a wide range of respiratory pathogens [15].

The M2e and the HA stalk influenza virus antigens were chosen for the development of a new probiotic influenza vaccine based on numerous data that indicated their broad protective potential. For example, a novel LAH (H7)-HBc virus-like particle vaccine has high potential as a vaccine candidate due to its high immunogenicity (both humoral and cell-mediated) and complete protection against heterologous type A influenza viruses (H7N9, H3N2 и H1N1) [10]. Another vaccine candidate, LAH (H3)-KLH, also provided complete protection from H3 subtype homologous viruses and in 60% of the cases conferred protection against heterologous H5N1 virus [17]. The fact that LAH-based constructs can elicit a broadly reactive immune response is intriguing. The absolute percent identity between subtypes for the amino acid 556–128 region of HA2 can be less than 50%; however, amino acids accessible to an antibody-binding site in the context of the helical structure may be more conserved [15]. Although our experiments involved only viruses belonging to the Group 1 HA, further studies will be set up to evaluate the ability of LAH-specific antibody induced by the new probiotic vaccine to confer protection against Group 2 HAs.

Another domain with high potential for developing broadly protective vaccines is the extracellular domain of the protein M2, M2e. Despite the location of M2e on the surface of influenza virions, it remains an extremely conservative region. Just as LAH, M2e cannot elicit a strong protective immune response on its own, therefore, various vectors for an M2e based vaccine have been evaluated, with most candidates demonstrating their immunogenic and protective efficacy [11]. The M2e-based vaccines are believed to induce a long-lasting M2e-specific antibody response [18], even though the anti-M2e antibodies are not neutralizing [19,20]. Owing to the conservation of the M2e epitopes among all influenza A viruses, and the clear ability of the recombinant *E. faecium* L3 strain encoding this antigen to induce protective M2e-specific, it is expected that the developed L3-LAH+4M2e live probiotic vaccine prototype will have a broad anti-influenza activity.

The use of the probiotic strain *E. faecium* L3 as a vector for the delivery of viral antigens and stimulation of immune responses on the mucous membranes has a number of advantages. The genus enterococcus are commensal bacteria and are considered as part of the intestinal microbiota of mammalian species [21]. It is now widely accepted that genetically modified probiotics are attractive carriers for the delivery of various antigens through the oral, gastrointestinal or nasal mucosa routes [22]. Immunization through the mucous membrane is one of the needle-free approaches that provide significant prophylactic and therapeutic effects due to the induction of a secretory and cellular immune response [23,24]. When probiotic bacteria enter the body by the oral route, they can propagate in the digestive tract and provide beneficial effects for the host [25,26].

Localization of viral antigens on the surface of probiotic bacteria may protect them from proteolysis, increase stability, and facilitate antigen presentation [27,28]. In clinical trials recombinant probiotic-based antiviral vaccines were safe for humans [29,30,31]. Probiotic-based vaccines for mucous membranes are easy to administer, are low in cost and provide durable T-cell memory responses. Plasma cells, after oral immunization, home to bone marrow and to effector sites in the lamina propria of the small and large intestine. The gut IgA response to oral vaccines is highly synchronized and strongly oligoclonal [32]. All these advantages make the *E. faecium* L3 a promising platform for universal influenza vaccine development.

Our study has a number of limitations. First, we did not generate *E. faecium* L3 recombinants expressing individual LAH or 4M2e antigens, therefore it was not possible to elucidate the precise role of each antigen in the observed anti-influenza protective effect. Second, only H1N1 influenza virus subtype was used in the challenge experiments, and further studies will be required to assess cross-protective potential of the live probiotic vaccine against evolutionary diverged influenza viruses. Third, we measured only the levels of antigen-binding circulating IgG antibody, and a deeper analysis of the induced immune responses to the inserted antigens will be needed, including analysis of antibody Fc-mediated functions, mucosal antibody responses, and the magnitude of circulating and tissue-resident memory T-cell responses. Finally, there were no direct comparisons of immunization with new probiotic vaccine prototypes and traditional vaccination approaches, including seasonal inactivated or live attenuated influenza vaccines. Such comparative studies would answer the question of whether the oral vaccination is beneficial in terms of providing improved cross-protection against various influenza A virus subtypes.

## 5. Conclusions

Overall, here we developed a new recombinant enterococcal strain, L3-LAH+4M2e, which proved immunogenic and protective in a mouse model and may be considered as a promising live probiotic vaccine candidate for influenza prevention. *E. faecium* L3 is a vector with high potential of effectively presenting various structural antigens to the host immune system. This live probiotic vaccine platform has the significant advantage that it is easily adaptable for production in existing dairy food processing facilities. In case of further successful pre-clinical and clinical trials, this vaccine has the potential to significantly improve vaccination coverage due to the ease of its manufacture and administration.

## Figures and Tables

**Figure 1 biomedicines-09-01515-f001:**
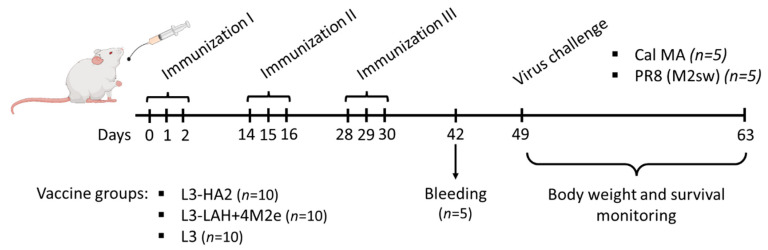
Overview of the mouse study design. BALB/c mice (*n* = 30) were immunized three times by feeding with a corresponding vaccine (L3-HA2, L3-LAH+4M2e or L3), two weeks apart. Serum samples were collected from retro-orbital sinus from five mice on Day 42 for immunological assessment. On Day 49, the mice were challenged with 3 LD_50_ of either mouse-adapted A/California/7/09 strain or a recombinant A/PR8-based virus carrying M gene of swine-origin lineage. Protection was assessed by monitoring weight loss and survival for 14 days post challenge.

**Figure 2 biomedicines-09-01515-f002:**
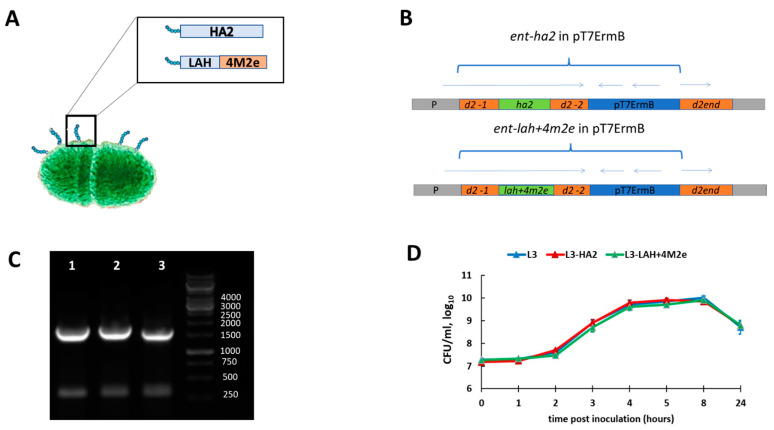
Generation and characterization of the live probiotic influenza vaccine candidates. (**A**) Schematic presentation of the chimeric *E. faecium* proteins. (**B**) Integration scheme of the plasmid pT7ermB with the ent-(lah+4m2e) into the chromosome of the strain *E. faecium* L3. P is the promoter of the gene d2; d2-1 is a region of the d2 gene encoding for N-(lah+4m2e) terminal part of D2 protein; d2-2 is a region of the d2 gene encoding for central portion of D2 protein; LAH and LAH+4M2e are DNA fragments encoding conserved influenza virus epitopes; d2-end is the end of the d2 gene encoding for the C terminus of D2 protein; pT7 ErmB is the integrative plasmid. Arrows correspond to the open reading frames in the integrated element. (**C**) Agarose gel electrophoresis of the amplified bacterial gene d2-1 containing inserts of influenza virus gene fragments: 1—L3-HA2; 2—L3-LAH+4M2e; 3—L3. (**D**) Growth properties of the *Enterococcus faecium* L3 strain and the enterococcal recombinants.

**Figure 3 biomedicines-09-01515-f003:**
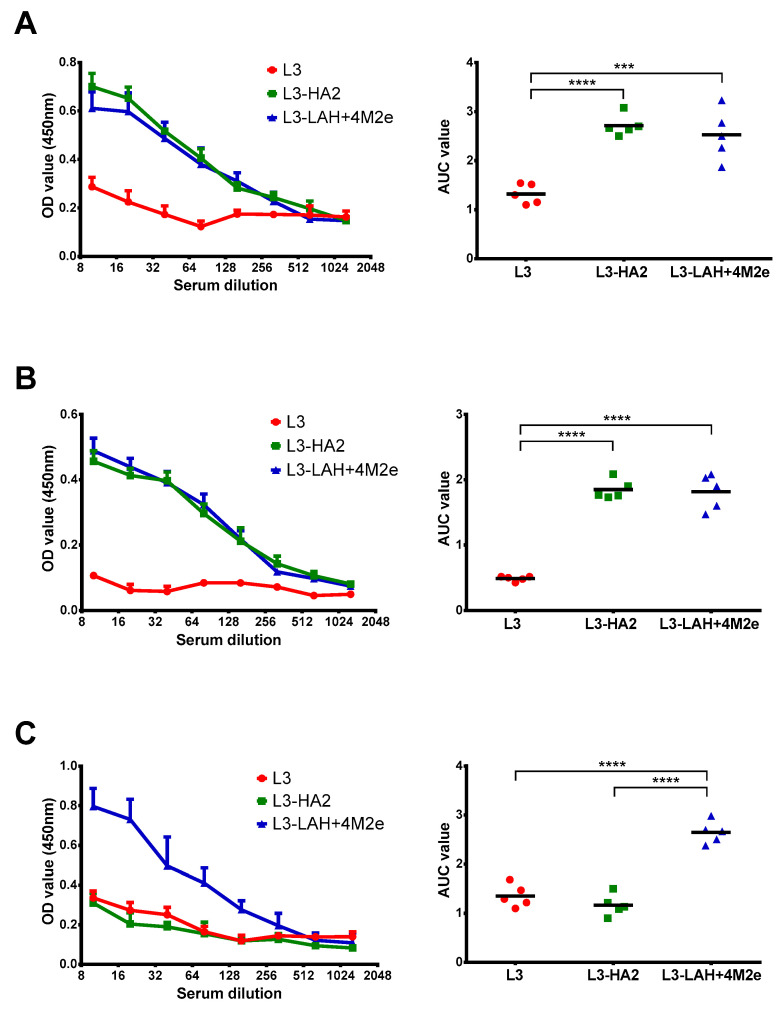
Serum IgG antibody immune responses after vaccination. Mouse sera were collected three weeks after third immunization. Sera were treated with L3-enterococci and then IgG antibody levels were assessed in ELISA against different influenza virus antigens: cH6/1 chimeric HA protein (**A**), LAH recombinant protein (**B**) and 3M2e recombinant protein (**C**). Left panel shows mean OD_450_ values for each serum dilution. Right panel shows the area under the OD_450_ curve values for each individual animal. Data were analyzed by one-way ANOVA with Tukey’s post-hoc multiple analyses test. ***—*p* < 0.001; ****—*p* < 0.0001.

**Figure 4 biomedicines-09-01515-f004:**
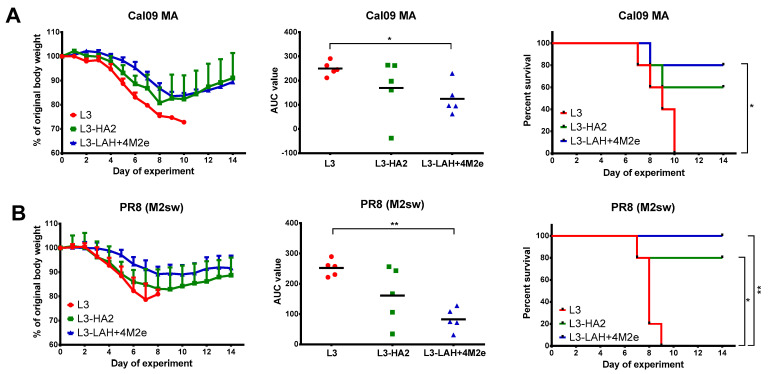
Protection of immunized mice against lethal influenza virus infection. Immunized mice were challenged with 3LD_50_ of either Cal09 MA (**A**) or PR8 (M2sw) (**B**) virus (*n* = 5 per virus). Body weight loss (left panel) and survival rates (right panel) were monitored for two weeks post-challenge. The AUC of body weight loss (middle panel) were compared by one-way ANOVA with Tukey’s post-hoc multiple analyses test. Survival rates were compared by Mantel–Cox log-rank test. *—*p* < 0.05; **—*p* < 0.01.

## Data Availability

The data presented in this study are available on request from the corresponding author.

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
