# Peer review of "A Live Probiotic Vaccine Prototype Based on Conserved Influenza a Virus Antigens Protect Mice against Lethal Influenza Virus Infection"

_biomedicines, 2021, doi:10.3390/biomedicines9111515_

Round 1
Reviewer 1 Report
Line 61. what do you mean with kindly provided? Please specify or remove
line 120. did administration perform using which via? please add
authors should clearly report how many mice were used in the intervention and in the control group.
I would suggest authors to refer to guidelines for reporting of in vivo study. Please refer to https://arriveguidelines.org/arrive-guidelines and mention it in the paper.
the rationale for selecting this type of control is missing. please add.
Please, consider to use a diagram for explaining the study design and type of intervention/control.
information on estimated sample size needed to estimate differences among group is not reported. Please add.
What about randomization or blinding?
Outcome variable, as well as all the other variables are not explained in the statistical section. please add.
lines 174-178 are methods and not results. Same also for lines 194-197. please revise.
Even if figures are very useful in showing results, they are not useful in providing data. My suggestion is also to add (eventually as supplementary material) tables showing the number behind the figures.
Discussion should be divided into: the first section is a summary of main results, then a discussion regarding the internal validity of the study, followed by external validity (comparison with previous studies) and lastly impact of the results.
The strengths and limitations of this study is not presented. Please add.
the impact and implication of these results should be added in the conclusions section.
In the funding section, please also state the role of funders.
Author Response
Reviewer 1.
We thank the reviewer for the careful evaluation of our work and all valuable suggestions.
Line 61. what do you mean with kindly provided? Please specify or remove
Authors’ response: we meant that this virus was shared by our colleague with no charge. We re-worded this sentence: “The mouse-adapted pandemic influenza virus strain A/California/7/2009 (H1N1) (Cal09 MA) was obtained from Smorodintsev Research Institute of Influenza (St. Petersburg, Russia)”.
line 120. did administration perform using which via? please add
Authors’ response: we added the word “oral” meaning that there was an oral administration using the mouse gavage needle (also see an additional figure for visualization of the route of vaccine administration).
authors should clearly report how many mice were used in the intervention and in the control group.
Authors’ response: we added the number of animals to the new Fig.1 (n=10 per group, 30 total).
I would suggest authors to refer to guidelines for reporting of in vivo study. Please refer to https://arriveguidelines.org/arrive-guidelines and mention it in the paper.
Authors’ response: we thank the reviewer for this suggestion. We tried our best to refer to these guidelines in reporting our findings.
the rationale for selecting this type of control is missing. please add.
Authors’ response: we added the following statement to the Methods section: “This type of control was required to exclude the non-specific antiviral effect of the enterococcal strain possibly mediated by the enterocin B peptide [16]”
Please, consider to use a diagram for explaining the study design and type of intervention/control.
Authors’ response: we added a figure with a mouse study design.
information on estimated sample size needed to estimate differences among group is not reported. Please add.
Authors’ response: we used a standard sample size for such kind of mouse studies. In accordance with the 3R's, studies should be designed to reduce the number of animals used to meet scientific objectives. For the sample size we used in this study (n=5) there was a power 78% to find the difference in the immunogenicity (i.e. AUC values 3 and 1.1) between study groups with an alpha 0.05 (G*Power version 3.0.10. software). This statement was added to the Methods section.
What about randomization or blinding?
Authors’ response: the animals used in the study are inbred and have an identical genetic background, therefore no specific randomization procedure was performed to divide them into study groups. The study also was not blinded.
Outcome variable, as well as all the other variables are not explained in the statistical section. please add.
Authors’ response: there were only three variables that were evaluated statistically: immunogenicity outcomes (AUC of OD450 values), pathological outcomes (AUC of weight loss) and survival rates. We added missing information about immunogenicity and pathology outcomes in the section 2.5.
lines 174-178 are methods and not results. Same also for lines 194-197. please revise.
Authors’ response: we thank the reviewer for this comment. The paragraph was corrected accordingly.
Even if figures are very useful in showing results, they are not useful in providing data. My suggestion is also to add (eventually as supplementary material) tables showing the number behind the figures.
Authors’ response: in such kind of research, it is usually not desirable to duplicate data in tables and figures, and we believe that additional numbers in the supplement will be redundant. In the study, we show individual AUC values for each sample, and this information is sufficient to understand the overall variation of the outcomes within each group.
Discussion should be divided into: the first section is a summary of main results, then a discussion regarding the internal validity of the study, followed by external validity (comparison with previous studies) and lastly impact of the results.
Authors’ response: we edited the discussion section to address this comment. Thus, the internal validity of the study was confirmed by the fact that the HA- and M2e-specific IgG antibodies were induced in those groups that contained corresponding antigens in the vaccine. Further we discuss previously published research in this field, comparing our results with those data. We also discuss possible mechanisms for the induction of immune responses after oral immunization with probiotic vaccines.
The strengths and limitations of this study is not presented. Please add.
Authors’ response: we added a paragraph about study limitations in the Discussion section.
the impact and implication of these results should be added in the conclusions section.
Authors’ response: we modified the Conclusions section accordingly.
In the funding section, please also state the role of funders.
Authors’ response: we added this section to the manuscript.
Reviewer 2 Report
The Authors have conducted an interesting and sound animal study to assess the feasibility of using Еnterococcus faecium L3 as a bacterial vector for oral immunization against influenza virus. They conclude that the recombinant enterococcal strain L3-LAH+4M2e can be con23 sidered as a promising live probiotic vaccine candidate for influenza prevention and warrants further evaluation.
I have only some very minor comments:
- some statements can be toned down (e.g., 'Impressive results have been obtained in studies of LAH sequences from various subtypes')
- can the Authors expand on the potential relevance of their findings for clinical practice?
- Can the Authors comment on the limitations of their study?
Author Response
The Authors have conducted an interesting and sound animal study to assess the feasibility of using Еnterococcus faecium L3 as a bacterial vector for oral immunization against influenza virus. They conclude that the recombinant enterococcal strain L3-LAH+4M2e can be considered as a promising live probiotic vaccine candidate for influenza prevention and warrants further evaluation.
Authors’ response: we thank the reviewer for the positive evaluation of our manuscript.
I have only some very minor comments:
some statements can be toned down (e.g., 'Impressive results have been obtained in studies of LAH sequences from various subtypes')
Authors’ response: we toned down this statement and several others.
can the Authors expand on the potential relevance of their findings for clinical practice?
Authors’ response: we added the following sentence to the Conclusions section to address this issue: “This live probiotic vaccine platform has the significant advantage that it is easily adaptable for production in existing dairy food processing facilities. In case of further successful pre-clinical and clinical trials, this vaccine has a potential to significantly improve vaccination coverage due to the ease of its manufacture and administration”
Can the Authors comment on the limitations of their study?
Authors’ response: this paragraph was added to the Discussion section.
Round 2
Reviewer 1 Report
- Regarding “the animals used in the study are inbred and have an identical genetic background, therefore no specific randomization procedure was performed to divide them into study groups. The study also was not blinded.”
- Please add this info in the text
- the first section is a summary of main results
- this point was not addressed by the authors
Author Response
1. Regarding “the animals used in the study are inbred and have an identical genetic background, therefore no specific randomization procedure was performed to divide them into study groups. The study also was not blinded.”
Please add this info in the text
Authors’ response: we added this statement to the Methods section.
2. the first section is a summary of main results
this point was not addressed by the authors
Authors’ response: we corrected the first paragraph of the Discussion section to highlight the main findings of the study.